# Iranian primary healthcare system's response to the COVID-19 pandemic using the healthcare incident command system

**Arezoo Yari[1,2], Homa Yousefi Khoshsabegheh[3,4], Yadolah Zarezadeh[2], Majid Amraei[4], Mohsen Soufi Boubakran[5], Mohamad Esmaeil Motlagh[6]***

**1** Department of Health in Emergencies and Disasters, School of Medicine, Kurdistan University of Medical Sciences, Sanandaj, Iran, **2** Social Determinants of Health Research Center, Research Institute for Health Development, Kurdistan University of Medical Sciences, Sanandaj, Iran, **3** Department of Health in Emergencies and Disasters, School of Public Health, Tehran University of Medical Sciences, Tehran, Iran, **4** Disaster Risk Management Office, Ministry of Health and Medical Education, Tehran, Iran, **5** Department of Mechanical Engineering, Urmia University, Urmia, Iran, **6** Department of Pediatrics, Ahvaz Jundishapur, University of Medical Sciences, Ahvaz, Iran

\* Dr.motlagh.ms@gmail.com

## Abstract

The present study aimed to evaluate the effects of the healthcare incident command system (HICS) on the district health networks (DHNs) covered by provincial Medical Universities (PMU) in terms of the management and commanding of the COVID-19 pandemic in Iran. This study was a cross-sectional survey. The study was performed in Iran in June 2020 in 60 DHNs, 41 of which had an active HICS. Data were collected on eight HCIS dimensions from all 60 DHNs by trained crisis management experts to evaluate the effects of HICS use on management of the COVID-19 pandemic. For all the 60 DHNs, the mean score of the COVID-19 incident command and management was 78.79 ± 11.90 (range 20–100); with mean scores highest for organizational support and coordination and lowest for logistic and planning. Significant differences were observed between the DHNs with active HICS and DHNs with inactive or no HICS in terms of the mean scores of incident management and command and their associated dimensions. According to the results, the HICS use had a positive impact on the improvement of incident management and command and all the related dimensions. Therefore, the HICS could be conducted and implemented in primary healthcare for the systematic and proper management of crises caused by infectious diseases and increasing primary healthcare system efficiency in response to these crises.

## Introduction

The COVID-19 pandemic as a public health emergency led to an increase in the mortality and morbidity rate around the world [1]. In other words, the COVID-19 pandemic is a mass casualty incident (MCI), requiring MCI disaster management based on the four main incident management stages of mitigation, planning, response, and recovery [2]. Spreading quickly across the world in 2019, the COVID-19 disease was declared as a global pandemic on March

**Data Availability Statement:** There are ethical restrictions on sharing the minimal data for this study. Data will be made available on request

through Kurdistan University of Medical Sciences Ethics Committee. Data be made available upon request to Kurdistan University of Medical Sciences Ethics Committee via email (ethiccommittee@muk. ac.ir) for researchers who meet the criteria for access to confidential data.

**Funding:** This study was financially supported by the Disaster Risk Management Offices at the Ministry of Health and Medical Education, Iran, in the form of a grant (no: 1399/2/21-14/7935) awarded to MEM. No additional external funding was received for this study. The funder had no role in study design, data collection and analysis, decision to publish, or preparation of the manuscript.

**Competing interests:** The authors have declared that no competing interests exist.

**Abbreviations:** DHC, District Health Centers; DHNs, district health networks; DRMO, Disaster Risk Management Offices; HICS, healthcare incident command system; MCI, mass casualty incident; MOHME, Ministry of Health and Medical Education; PHC, primary health care; PMU, provincial Medical Universities.

11th, 2020 [3,4]. It has been considered as an unprecedented challenge for health systems [5]. An important issue in this regard is the increased number of the patients, requiring systems to become more operationally compatible with the new situation [6]. The current pandemic led to a shortage of ventilators and other medical equipment needed by the health-care system [7]. Consequently, health systems are under tremendous pressure to protect lives during this crisis [8]. Meanwhile, the disease has significantly impacted the health systems of low- and middle-income countries [9]. Iran confirmed the spread of SARS-CoV-2-virus in the country on February 19th, 2019; afterwards, all the provinces reported cases of the COVID-19 disease up to March 5th, 2020 due to the rapid spread of the virus [10]. Furthermore, an increase was also reported in the load of the disease and number of the COVID-19-related deaths in Iran due to concomitance with the international sanctions imposed on this country [3], as well as the shortage of medical, pharmaceutical, and laboratory equipment [11].

The COVID-19 pandemic showed that the world is constantly affected by outbreaks of various diseases despite the advances in medical sciences [12]. In the current pandemic, various technological solutions have been developed to reduce the effects of the disease in the world. Such examples are major advances in biosensor-based technological solutions for COVID-19 diagnosis [4], advances in tissue engineering to support the treatment of patients with COVID-19 [13], as well as bioengineering technology to tackle the COVID-19 pandemic [7]. However, Medical sciences will fail to manage this crisis without the principles of crisis management. Emergency management is an inherent element of health systems [14], and emergency planning is a key responsibility of health systems [15]. Based on global experiences, the healthcare incident command system (HICS) is a valuable incident management tool, playing a pivotal role in increasing the quality of primary healthcare services [16]. According to the crisis management cycle, the HICS activation in the phase of response to the COVID-19 pandemic is essential to the practical organization of the affected areas by the viral infection [6]. The structure is operated by using a logical and integrated management structure, describing the responsibilities, developing clear reporting channels, and applying a common and simple terminology system for the further coordination of the institutions involved in unexpected events [17,18]. For instance, using the health information system technology during the COVID-19 pandemic has improved information management and patient treatment [19]. Moreover, the method could be used by public healthcare centers to enhance emergency planning and response in disasters and emergencies regardless of their size or ability in patient care [20]. HICS consists of five operational systems, including command, planning, operation, logistics, and finance/administration [21].

Although the COVID-19 pandemic is regarded as a health emergency, it cannot be combated only through normal medical and preventive measures. The most essential components of contagious disease management in emergencies are the presence of a healthcare system and rapid introduction of proper control measures. In fact, the effective management of the crisis requires health interventions through public health surveillance systems [22,23]. The establishment of a surveillance system in emergencies, incidents or disasters is a complicated process and requires various resources, such as human resources and specific administrative facilities and equipment [24].

The HICS provides the operational coordination required by an organization to respond to disease outbreaks [6]. In addition, the organization defines the roles and responsibilities of individuals, organizational response functions, power lines, as well as communication and management practices [25]. Today, the HICS is widely used by healthcare systems [5]. Iran has a large network of primary healthcare (PHC), providing healthcare services to the rural and urban areas of the country [26]. Based on the national disaster response framework, the Ministry of Health and Medical Education (MOHME) has encouraged that all health systems to use

HICS as an essential infrastructure since 2011 [27]. With the outbreak of the SARS-CoV-2-virus in Iran, the authorities officially announced the activation of HICS in the healthcare system on all levels.

In response to the COVID-19 Pandemic threat, the UW Medicine Systems in the health system of the United States used the HICS instructions to collect organizational resources [5]. However, a small number of studies have been published on the implementation and evaluation of this framework despite its long-term use in the world [16,28]; in other words, it seems no comprehensive research has investigated the positive and negative impacts of this system [28]. Overlooking some of the quality-related aspects of the issue might be due to the importance of establishing the HICS for emergencies [16].

Given the outbreak of the SARS-CoV-2-virus in Iran and history of using the HICS in the primary healthcare system of the country, the present study aimed to evaluate the effects of using the HICS in the district health networks (DHNs) covered by provincial medical universities (PMUs).

## Methods

### Study area

This cross-sectional study was conducted in the DHNs covered by the PMUs in Iran in June 2020. In total, 60 out of 62 DHNs participated in this research. In Iran, public health care (PHC) is based on the DHN; in other words, every area has a regional DHN, which is an organization constituting of District Health Centers (DHC), rural health centers, urban health centers, health houses, and health posts that provide healthcare services to all parts of the region. In addition, the network is considered to be a sub-organization of the PMU, each PMU covers one or more regions and monitors PHC provision in the province. Notably, some provinces have more than one PMU and the regions of the province have been divided between them [29] (Fig 1).

In the Iranian primary healthcare system, the HICS has a command group and four section including planning, operation, logistics, and finance/administration. The command group encompasses the five situations of incident commander, public information officer, liaison officer, safety officer, and security officer [30] (Fig 2).

Iranian DHNs play a key role in disease prevention and dissemination during the COVID-19 pandemic through a wide range of primary health activities, the most important of which are participation in making decisions related to social and health issues, training and monitoring health measures, community screening, referral and testing measures of the patients, case surveillance and follow up of the patients. In addition, DHNs participates in providing personal protective equipment (PPE) to health personnel and vulnerable community groups. This system delivers ongoing primary healthcare services to the communities.

### Data collection

In the present study, we applied the HICS evaluation tool to assess the HICS in Iranian DHNs during the COVID-19 pandemic. The tool consisted of 50 items to assess incident management in the four dimensions of organizational support, the implementation process of incident management, resource and equipment management, and human resource management. The executive process of incident management is performed through coordination, communication, and operations.

This tool has been designed and finalized with a completely scientific method after collecting all aspects affecting the functioning of the HICS, as well as the factors that should be considered in evaluation of the HICS. In fact, these factors were extracted through two studies

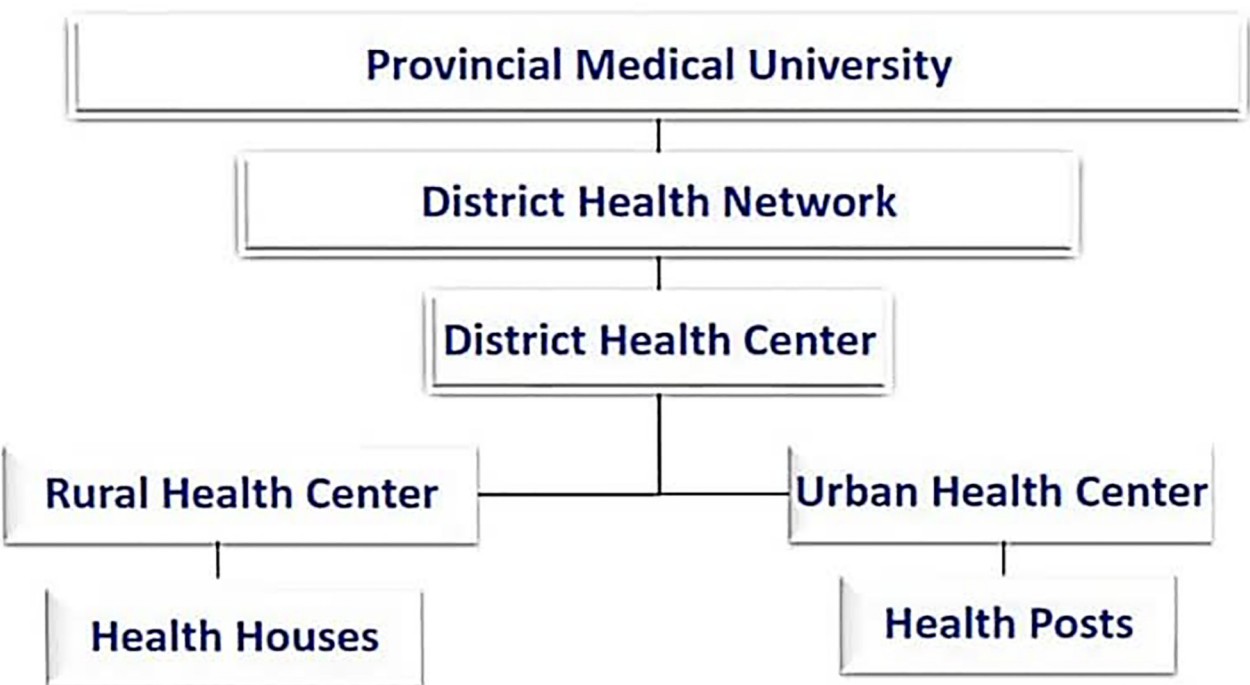

**Fig 1. Public healthcare system structure in Iran.** This figure was constructed by the authors using existing information in the Iranian Ministry of Health.

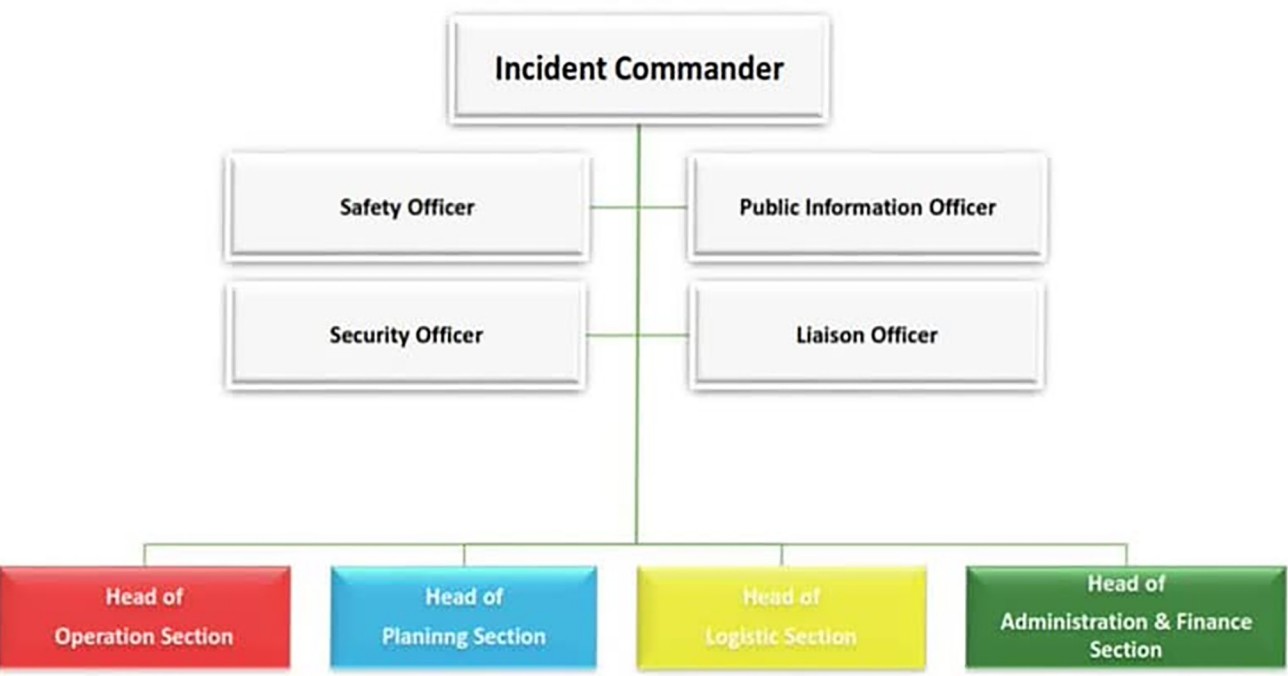

**Fig 2. Healthcare incident command system in Iran.** This figure was constructed by the authors using existing information in the Disaster Risk Management Office in Iranian Ministry of Health. Reference number 21 was used to construct this figure.

including: a systematic review [31] during 1990 to 2017 and one qualitative study in Iran [32]. Based on the factors identified in the two mentioned studies, the HICS evaluation tool was designed and psychometrically evaluated.

The validity and reliability of the HICS evaluation tool has been confirmed for the Iranian population. The validity of this tool used was evaluated by factor analysis using a correlation matrix. At this stage, the Kaiser-Meyer- Olkin (KMO) was estimated at 0.975, and the Bartlett's test was also significant. The reliability of the instrument was confirmed at the Cronbach's alpha of 0.997 [32].

All DHNs in Iran have Disaster Risk Management Offices (DRMO) with disaster risk management experts experienced in attending the training courses of disaster management and HICS. In each DHN selected in the current research, data was collected by the disaster risk management expert of the DHN. Information was collected from the DHN executives or key members of the COVID-19 Crisis Committee, each of whom was partly responsible for the COVID-19 crisis management. For instance, the questions regarding the human resources were asked of the human resources department manager, and the questions regarding the equipment and resources were asked of the head of the equipment and resources department. The surveyors were trained on the collection of data online and via virtual sessions and allowed to contact the research team in case of ambiguities. After the necessary training and providing explanations about the objectives and methodology of the research and ethical considerations to ensure the strength of the collected data, the research team meticulously monitored the performance of the surveyors. After data collection, post-performance control, assessment, and statistical control of the tools were performed as well.

## Data analysis

In the present study, the total scores of incident command and its dimensions were determined by calculating the score of each dimension separately and weighting the score based on the number of the items in each dimension. Consequently, as the weight of all dimensions is not the same and the weight of the dimensions is based on the number of items in each dimension, the scores of all items in all dimensions were summed up and divided by the total number of the items to calculate the total score of incident command. The total scores of incident command and its dimensions were determined within the range of 20–100. These scores were classified as low, medium, and high. The scores below the first quarter were defined as low, the scores within the quarters one to three were considered medium, and the scores above the third quarter were defined as high. Therefore, the scores of the quarters in each dimension were different (S1 Table). In addition to the above mentioned, the Iranian DHNs were divided into two groups; the first group had an active HICS, and the second group had no HICS or just had an inactive HICS. Meanwhile the DHNs that had HICS and now their HICS is inactive, have same function with DHNs without HICS, these two were considered as one group. Both groups completed the questionnaire of the COVID-19 crisis management status, and the results were compared. Finally, the impact of the HICS activation on the improvement of the response rate to the COVID-19 crisis in the country was assessed by comparing various aspects of incident management in both groups (Fig 1).

Data analysis was performed in SPSS version 22 using descriptive indexes (percentage, mean, and standard deviation) and t-test.

## Ethics approval and informed consent to participate

This study has obtained the approval of Kurdistan University of Medical Sciences' Institutional Review Board (IRB). The IRB follows the stipulated clauses of the Helsinki Declaration. Ethics

Committee of Kurdistan University of Medical Sciences approved this study and Ethical Approval Number "IR.MUK.REC.1399.020" was obtained. This committee follows the required principles of the Helsinki Declaration. Informed consent was obtained and ethical principles such as confidentiality and the right of withdrawal at any time was explained to the participants. The participants in this study were previously informed about the characteristics of the study. They were all asked to participate in the study and to provide written consent to confirm the participation in the study.

## Results

In the present study, 98% of the DHNs cooperated with the researchers. In total among the 60 participating DHNs, 68.3% (n = 41) had activated their HICS, while 31.7% had no HICS or had not any active HICS (n = 19). The total mean score of the COVID-19 crisis command and management based on various dimensions of the framework was 78.79±11.90 (score range: 20–100) in all the Iranian DHNs. Based on the score classification, 25% of the DHNs obtained high scores, while 50% and 25% obtained medium and low scores, respectively. The findings also indicated that the DHNs had the highest score in terms of organizational support and coordination and the lowest score in terms of logistic and planning. In other words, the PMUs with an active HICS had better planning and communication scores, while the DHNs with an inactive or no HICS had a low or medium planning and communication scores. Moreover, the DHNs with an active HICS system had more score in favorable organizational support compared to the DHNs without an HICS or those that had not activated the HICS (Table 1).

Based on the classification of the incident management and command and the associated dimensions, the obtained results showed that 34.2% of the DHNs with active HICS commanded and managed the COVID-19 crisis at a high level, while 73.7% of the DHNs with no active HICS commanded and managed the COVID-19 crisis at a low level. In addition, the DHNs with active HICS achieved higher scores in the HICS dimensions compared to the DHNs with inactive HICS, so that none of the DHNs with inactive HICS could be considered high-level in terms of planning, logistics, and human resource management. Further, in other dimensions including organizational support, operational, coordination and communication, few of the DHNs with no active HICS were in the high-level, while about half of the DHNs with active HICS were placed in the high or medium levels in these dimensions (Table 2).

The analytical results of the present study indicated that the mean scores of command and incident management and the associated dimensions significantly differed between the DHNs with an activated HICS and those with an inactive HICS (Table 3). These dimensions included the following: organizational support, operational, coordination, communication, planning, logistic, and human resource management.

**Table 1. Healthcare incident command level based on the HICS dimensions at time of COVID-19 in the Iranian DHNs (n = 60).**

| Healthcare Incident Command system dimensions | Low n (%) | Average n (%) | High n (%) | Total Score Mean ± SD |
|---|---|---|---|---|
| Total Incident Management and Command | 15(25.0) | 30(50.0) | 15(25.0) | 78.79±11.90 |
| Organizational support | 26(43.3) | 10(16.7) | 24(40.0) | 84.33±11.63 |
| Operational | 12(20.0) | 31(51.7) | 17(28.3) | 75.77±12.17 |
| Coordination | 16(26.7) | 30(50.0) | 14(23.3) | 82.66±14.54 |
| Communication | 17(28.3) | 28(46.7) | 15(25.0) | 79.86±13.97 |
| Planning | 17(28.3) | 30(50.0) | 13(21.7) | 73.11±16.14 |
| Logistic | 15(25.0) | 35(58.3) | 10(16.7) | 69.13±17.24 |
| Human resource management | 30(50.0) | 16(26.7) | 14(23.3) | 81.91±14.73 |

**Table 2. Comparison of DHNs with an active HICS and without active HICS or no HICS according to HICS' dimensions at time of COVID-19 in Iran (n = 60).**

| Healthcare Incident Command system dimensions | DHNs with an active HICS (n = 41) | | | DHNs without active HICS or no HICS (n = 19) | | |
|---|---|---|---|---|---|---|
| | Low, n (%) | Average n (%) | High, n (%) | Low, n (%) | Average, n (%) | High, n (%) |
| Total Incident Management and Command | 5(12.2) | 22(53.7) | 14(34.1) | 10(52.6) | 8(42.1) | 1(5.3) |
| Organizational support | 12(29.3) | 7(17.1) | 22(53.7) | 14(73.7) | 3(15.8) | 2(10.5) |
| Operational | 6(14.6) | 20(48.8) | 15(36.6) | 6(31.6) | 11(57.9) | 2(10.5) |
| Coordination | 5(12.2) | 23(56.1) | 13(31.7) | 11(57.9) | 7(36.8) | 1(5.3) |
| Communication | 10(24.4) | 19(46.3) | 12(29.3) | 7(36.8) | 9(47.4) | 3(15.8) |
| Planning | 7(17.1) | 21(51.2) | 13(31.7) | 10(52.6) | 9(47.4) | 0(0.0) |
| Logistic | 7(17.1) | 24(58.5) | 10(24.4) | 8(42.1) | 11(57.9) | 0(0.0) |
| Human resource management | 14(34.1) | 13(31.7) | 14(34.1) | 16(84.2) | 3(15.8) | 0(0.0) |

**Table 3. Comparison of mean scores of HICS dimensions at time of COVID-19 in the Iranian DHNs (n = 60).**

| Healthcare Incident Command system dimensions | DHNs with an active HICS (n = 41), Mean± SD | DHNs without active HICS or no HICS (n = 19), Mean± SD | t | P value |
|---|---|---|---|---|
| Total Incident Command | 83.02±9.35 | 69.67±11.88 | 4.710 | 0.000 |
| Organizational support | 87.80±9.98 | 76.84±11.62 | 3.753 | 0.000 |
| Operational | 78.69±11.47 | 69.47±11.50 | 2.895 | 0.005 |
| Coordination | 87.51±12.03 | 72.21±14.21 | 4.322 | 0.000 |
| Communication | 83.02±12.28 | 73.05±15.25 | 2.706 | 0.009 |
| Planning | 79.67±12.64 | 58.94±13.74 | 5.747 | 0.000 |
| Logistic | 74.82±13.97 | 56.84±17.51 | 4.274 | 0.000 |
| Human resource management | 87.43±10.06 | 70.00±16.32 | 5.085 | 0.000 |

## Discussion

Several factors affect COVID-19 crisis management in health systems. However, few studies regarding the COVID-19 crisis have focused on the management of the crisis based on disaster management principles. A key strength of our study was the evaluation of various dimensions of COVID-19 crisis command based on a global algorithm, which encompassed all the aspects of incident command and management in an emergency. In addition, this study is effective in improving our understanding of the role of PHC systems in controlling the COVID-19 pandemic.

Investments and PHC systems strengthening as an inherent element of the health system play a pivotal role in the management and control of the COVID-19 crisis [33]. According to this research, the DHNs with an active HICS system had more score in favorable organizational support compared to the DHNs without an HICS or those that had not activated the HICS. Overall, an HICS has multiple functional areas with specific purposes. The goals of each region are hierarchically in line with the basic goals of the system and programs set by the upper levels, so that the presence of HICS functional areas could ultimately enable the organization to achieve multiple goals [34], which in turn improved organizational support.

The value of the HICS system in the reduction of organizational differences has been raised as an issue especially in large-scaled events [35]. This could be due to the fact that the mere existence of an HICS does not guarantee organizational success and may not enhance organizational support. In the COVID-19 pandemic, the reorganization of PHC services and

strengthening the leadership role may be vital in providing routine PHC activities, meeting the current healthcare needs, and controlling the pandemic [36]. In fact, the strong commitment of organizational executive leaders, culture of organizational promptness, budget to support planning, training, and practice significantly impact executive commitment and the success of HICS, which in turn promote organizational support [28]. Furthermore, determining the organizational hierarchy of the HICS structure based on application requirements rather than the titles and positions of individuals plays a crucial role in organizational success [37].

The HICS is an accurate, broad-spectrum management system in public health environments to define operational specifications, interactive management components, and incident management structures [34]. This framework is also a standard system for disaster response [38], as well as a standard management system [39]. Nevertheless, Stephen S. Morse has claimed that many health organizations are not comfortable with the use of this system [40]. In another study, Burkle F. M. et al. discussed the application of the framework in health-related disasters on a large scale [41]. In this regard, our findings indicated that the executive operation score of the DHNs with an active HICS was significantly higher compared to that of DHNs with an inactive HICS or no HICS during the COVID-19 crisis. In fact, the HICS has been observed to improve organizational operations during disasters through prioritizing operation checklists, identifying positions, and teamwork [20]. In addition, the framework is considered proper for complicated and multidisciplinary operations [42], so that the failure of one component of the HICS would not lead to the failure of the other components [43]. The efficiency of the system and its operations could be enhanced by measures such as promoting familiarity with the HICS [17], and the organizational structure of the HICS [44], in addition to the scientific and practical training of staff regarding the framework [45]. The HICS provides a standard response format and increases coordination since it could be recognized by the organizations and various sectors that are responsible for incident management, thereby affecting the response rate [20].

From the perspective of primary health care, an intersectoral and coordinated response is critical to the effective control of the COVID-19 crisis [46]. According to the results of this study, the HICS could improve the coordination status in the COVID-19 crisis management. In fact, the framework provides the opportunity for the coordinated response to emergencies [42,43,47,48], as well as the multidisciplinary coordination for response to public health threats [42]. In addition, the system creates unity in severe crises by enabling the use of reciprocal efforts [35,49].

In the current research, the DHNs with an active HICS had better communication score in the COVID-19 crisis management. The difference between the DHN groups was considered significant in this regard. In general, the HICS is a communication system [50], which enables the communication and sharing of resources among health institutions and organizations [47]. The framework also promotes administrative communication [28] through establishing clear inter-organizational associations [48] and preventing unnecessary communications [51], facilitating communication among hospitals, medical emergencies, other response systems [50,52] and other organizations [53]. In the current pandemic and in situations such as the national lockdowns, using the telemedicine technology in health systems could prevent unnecessary meetings and solve the problem of social distancing to a great extent [54]. For instance, using other new technologies, such as the Chinese Blockchain technology, in the current pandemic has been reported to be highly beneficial [55]. Therefore, the simultaneous use of these technologies and the HICS in the health system could remarkably improve communication in the COVID-19 crisis.

Although HICS is not a disaster plan, it acts as a guide to the proper organizational management of an emergency [47,50], thereby increasing the ability to retrieve scheduled and

unplanned events [56]. It remarkably facilitates responding, planning, decision-making, and documentation [18]. To this end, new technologies (e.g., Industry 4.0) could be employed to plan and implement COVID-19 -related activities, as the positive effects and applications of Industry 4.0 technologies in health systems have been demonstrated in the current pandemic [57].

In this study, among the studied dimensions, the lowest command scores in the DHNs after logistic dimension belonged to the planning dimension. Considering that the HICS improves planning, it is essential to assess its activation and correction in organizational planning prior to an incident or disaster [28]. Moreover, the framework could help eliminate the lack of the executive commitment of the system in the planning department [28]. Measures such as effective planning, communication, and intersectoral coordination among managers, and policymakers in the field of PHC could result in upgraded and enriched overall preparedness [58].

The current pandemic has brought about economic problems for the health sectors worldwide. Consequently, the optimal use of the capacity of the PHC structure has been hindered [59]. According to our results, the weakest incident command dimension in the Iranian DHNs was the logistics dimension. This might be due to the sanctions against Iran during the current pandemic [3]. In addition, the effects of the COVID-19 pandemic on the supply chain of medical equipment and the resulting shortages in the world have exacerbated this problem [7]. Considering this issue, the elimination of the financial barriers of the health system and financially supporting the establishment of the HICS improve the efficiency of the framework [60]. However, it could be concluded that the system would be able to reduce the loads of logistic problems during crises by using all the resources for problem-solving [37], supply needs, resources, and equipment [61], thereby providing the health facilities required for emergency management [39].

Human resources in the field of PHC have a pivotal role in responding to the COIVD-19 pandemic and preventing mortality and morbidity due to the disruption or delay in the access to primary healthcare services. Meanwhile, some reports suggest that the PHC workforce has suffered tremendous pressure in the current pandemic [62]. According to our results, the human resource management of the DHNs with an active HICS was significantly better than the DHNs with an inactive or no HICS. By providing services in the shortest possible time [17], identifying the areas for expanding services during patient overload periods [61], preventing rework [48], and using human resources regularly [49], the HICS encourages individuals to take on the necessary responsibilities in a timely manner [60]. This in turn results in effective human resource management, as well as adequate and efficient human resources [48].

The main limitation of this study is related to its design. We did not investigate any causal relations, in fact, we merely assessed the associations between HICS and COVID-19 crisis management in Iranian Primary Healthcare Systems. This means that the results cannot be necessarily and solely attributed to the HICS rather, other organizational or contextual factors, which have not been considered in our study, may have contributed to the results.

## Conclusions

This study aimed to evaluate the effects of HICS on the DHNs covered by the PMUs in terms of the management and commanding of the COVID-19 pandemic in Iran. According to the results, it seems that activating the HICS and its dimensions is associated with improvement in the management and command of the COVID-19 crisis. Thus, policymakers and health managers are advised to identify the HICS challenges and complications in organizational

planning. Similarly, they should conduct exercises and maneuvers according to the principles of the HICS. Moreover, In order to use the HICS in the healthcare networks, primary healthcare systems should activate the HICS and apply principles in the phase of responding to crises such as COVID-19 pandemic.

In conclusion, the use of HICS seems to be associated with systematic management and proper command of infectious disease crises. Therefore, it may increase the efficiency of primary healthcare systems in response to these crises.

## Supporting information

**S1 Table. Dimension scoring of the HICS evaluation tool.**
(DOCX)

**S1 File. Questionnaire to evaluate primary healthcare (PHC) response to the COVID-19 pandemic on the district health networks (DHNs) with active HICS.**
(DOCX)

**S2 File. Questionnaire to evaluate primary healthcare (PHC) response to the COVID-19 pandemic on the district health networks (DHNs) with inactive or no HICS.**
(DOCX)

## Acknowledgments

This study was carried out with the support of the Kurdistan University of Medical Sciences and Disaster Risk Management Office in Ministry of Health. We thank Golnaz Babamiri and Zahra Motlagh for improving the paper. Moreover, the authors would like to thank all participants of the study who gave us their precious time.

## Author Contributions

**Conceptualization:** Arezoo Yari, Mohsen Soufi Boubakran, Mohamad Esmaeil Motlagh.

**Data curation:** Majid Amraei.

**Formal analysis:** Arezoo Yari, Majid Amraei.

**Funding acquisition:** Arezoo Yari.

**Investigation:** Arezoo Yari, Homa Yousefi Khoshsabegheh, Majid Amraei.

**Methodology:** Arezoo Yari.

**Project administration:** Mohamad Esmaeil Motlagh.

**Resources:** Homa Yousefi Khoshsabegheh.

**Software:** Arezoo Yari, Mohsen Soufi Boubakran.

**Validation:** Mohsen Soufi Boubakran.

**Visualization:** Mohsen Soufi Boubakran.

**Writing – original draft:** Arezoo Yari, Yadolah Zarezadeh.

**Writing – review & editing:** Arezoo Yari, Yadolah Zarezadeh.

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
