## [Decision Letter · Decision Letter 0]

7 Feb 2023

PONE-D-22-30776Iranian Primary Healthcare System's Response to COVID-19Pandemic using Healthcare Incident Command SystemPLOS ONE

Dear Dr. Motlagh,

Thank you for submitting your manuscript to PLOS ONE. After careful consideration, we feel that it has merit but does not fully meet PLOS ONE’s publication criteria as it currently stands. Therefore, we invite you to submit a revised version of the manuscript that addresses the points raised during the review process.

Please make the changes as suggested by the reviewer.

Revise the abstract and other sections

We look forward to receiving your revised manuscript.

Kind regards,

Rabia Hussain, PhD

Academic Editor

PLOS ONE

Journal Requirements:

2. During your revisions, please note that a simple title correction is required: There is a missing space in the title "COVID-19Pandemic", please correct this in the online submission information."

"This study was carried out with the support of the Kurdistan University of Medical Sciences and funded by Disaster Risk Management Office in Ministry of Health."

7. In your Data Availability statement, you have not specified where the minimal data set underlying the results described in your manuscript can be found. PLOS defines a study's minimal data set as the underlying data used to reach the conclusions drawn in the manuscript and any additional data required to replicate the reported study findings in their entirety. All PLOS journals require that the minimal data set be made fully available. For more information about our data policy, please see http://journals.plos.org/plosone/s/data-availability.

Additional Editor Comments:

Dear Authors,

Please revise your manuscript as suggested by the reviewers.

Reviewers' comments:

Reviewer's Responses to Questions

**Comments to the Author**

1. Is the manuscript technically sound, and do the data support the conclusions?

Reviewer #1: Yes

Reviewer #2: No

2. Has the statistical analysis been performed appropriately and rigorously? 

Reviewer #1: Yes

Reviewer #2: I Don't Know

3. Have the authors made all data underlying the findings in their manuscript fully available?

Reviewer #1: No

Reviewer #2: No

4. Is the manuscript presented in an intelligible fashion and written in standard English?

Reviewer #1: Yes

Reviewer #2: Yes

5. Review Comments to the Author

Reviewer #1: “Iranian Primary Healthcare System’s Response to COVID-19Pandemic

using Healthcare Incident Command System”

Comments on the paper by Arezoo Yari et al

submitted to PLOS ONE (MS number PONE-D-22-30776)

Author : P.N. Lee

Date : 23rd November 2022

While the paper is generally well written and the results clear and well analysed, there are a few points that demand some attention. These are considered below in the order they appear in the paper and not in order of importance.

1. The full title would be better if there were a space between “COVID-19” and “Pandemic” and if the word “the” was inserted after “Response to” and after “using”.

2. I found the Methods section of the Abstract somewhat confusing at first, as it stated that some DHNs had an active HICS and some did not, so how could data on HICS be collected from those that did not have an active HICS? Later on in the paper it appears that data were in fact collected from all the 60 DHNs. It might also be a good idea to mention the number of DHNs with active and inactive HICS. Perhaps the Method section might read something like the following:

“Methods: The study was performed in Iran in June 2020 in 60 DHNs, 41 of which had an active HICS. Data were collected on eight HCIS constructs from all 60 DHNs by trained crisis management experts to evaluate the effects of HICS use on management of the COVID-19 pandemic”.

3. The results might better start:

“For all the 60 DHNs, the mean score of the COVID-19 incident command and management was 78.79 ± 11.90 (range 20-100); with mean scores highest for organizational support and cooordination and lowest for logistic and planning”.

4. While it is clear enough from Tables 2 and 3 that scores on total incident command and the other seven dimensions were clearly higher in DHNs with an active HICS (p<0.01 in all cases), I wonder whether it is enough to infer from this that HICS should always be active. The study does not investigate health endpoints, and it would be interesting to know, for example, whether the 41 DHNs with active HICS had a lower mortality rate among patients admitted for COVID-19 than did the 19 DHNs without active HICS. Does having active HICS work as regards the virus?

5. Near the end of the Introduction section, there is a statement that MOHME “necessitated that all health systems use HICS as an essential infrastructure since 2011” and that the following the outbreak of SARS-COV-2 in Iran, “the authorities officially announced the activation of HICS in the healthcare system on all levels”. That being the case, how did it come about that 19 of the 60 DHNs did not use the HICS?

6. Under “data analysis” it is made clear that the classification of total incident management and command into low, average and high was based on putting the bottom 25% as low, the top 25% as high and the rest as average. But there is nothing about how low, average and high were defined for the seven constructs from organizational support to human resource management. It would be useful to give fuller details as to how these scores were derived, perhaps as a supplementary file which details the questions actually asked and the possible replies.

7. In the Discussion section there is a paragraph starting “In this study, among the studied dimensions, the lowest command scores in the DHNs belonged to the planning dimension”. But according to Table 1, the lowest score is for the logistic dimension.

8. While the paper is well written, the last paragraph of the introduction could be improved by replacing “didn’t” by “did not”, and by inserting “been” before “considered in our study”.

Reviewer #2: Dear Authors

This is good effort to explain Iranian Primary Healthcare System's Response to COVID-19Pandemic using Healthcare Incident Command System.

It is well written .

But in my opinion it is more useful for local Iranian Journal.

6. PLOS authors have the option to publish the peer review history of their article (what does this mean?). If published, this will include your full peer review and any attached files.

Reviewer #1: **Yes: **Peter N Lee

Reviewer #2: No

---

## [Author Response · Author response to Decision Letter 0]

8 May 2023

PLOS ONE JOURNAL 

Dear Editor 

Thank you very much for considering our manuscript for publication. We appreciate the comments made by the respected the reviewers. 

We carefully read the comments and considered the comments and made changes and /or made our manuscript clearer accordingly. 

With respect to the opinions of the reviewers and editors, the article was revised. Please note that all changes are done in the main text. 

Journal Requirements:

 and https://journals.plos.org/plosone/s/file?id=ba62/PLOSOne_formatting_sample_title_authors_affiliations.pdf

Response: Thanks for your comment. Based on your comment we revised our article based on The PLOS ONE style templates.

2. During your revisions, please note that a simple title correction is required: There is a missing space in the title "COVID-19Pandemic", please correct this in the online submission information."

Response: Thanks for your comment. Based on your comment we made this correction on the title and added the space in the tittle "COVID-19 Pandemic", in the online submission information.

"This study was carried out with the support of the Kurdistan University of Medical Sciences and funded by Disaster Risk Management Office in Ministry of Health."

Response: Thanks for your comment. Based on your comment we revised the Acknowledgments Section of my manuscript and removed any funding-related text from the manuscript. Also, we have added and highlighted my amended statements (related to the funding section) in the cover letter, we hope this section will be added to the online version. 

Response: Thanks for your comment. Based on your comment we moved our ethics statement to the method section and deleted it from last section in the article. these changes are highlighted in yellow color in the last part of methods section, page 9, lines 6-15.

Response: Thanks for your comment. Based on your comment we have added and highlighted my amended statements (related to the funding section) in the cover letter, we hope this section will be added to the online version. 

Response: Thanks for your comment. Based on your comment we added an ORCID iD for the corresponding author Editorial Manager account.

7. In your Data Availability statement, you have not specified where the minimal data set underlying the results described in your manuscript can be found. PLOS defines a study's minimal data set as the underlying data used to reach the conclusions drawn in the manuscript and any additional data required to replicate the reported study findings in their entirety. All PLOS journals require that the minimal data set be made fully available. For more information about our data policy, please see http://journals.plos.org/plosone/s/data-availability.

Response: Thanks for your comment. Based on your comment we update and added the data availability statement section with the name and contact details of the local ethics committee upon our cover letter. the data availability statement section is shown in red color and highlighted with yellow color in cover letter. 

Additional Editor Comments:

Dear Authors,

Please revise your manuscript as suggested by the reviewers.

Reviewers' comments:

Reviewer's Responses to Questions

Response: Thanks for your comment. Based on your comment we revised our manuscript as suggested by the reviewers. Also, we did not edit the HTML markup suggested by you. 

Comments to the Author

1. Is the manuscript technically sound, and do the data support the conclusions?

Reviewer #1: Yes

Reviewer #2: No

Responses: Thanks for your comment. In this cross-sectional study, we tried to carry out all the stages of implementation technically according to the principles of a cross-sectional study.

2. Has the statistical analysis been performed appropriately and rigorously?

Reviewer #1: Yes

Reviewer #2: I Don't Know

Responses: Thanks for your comment. All statistical analysis were performed in consultation with one of the authors who has expertise in statistics.

3. Have the authors made all data underlying the findings in their manuscript fully available?

Reviewer #1: No

Reviewer #2: No

Responses: Thanks for your comment. we cannot share our data publicly but the data will be made available on request through Kurdistan University of Medical Sciences Ethics Committee, to researchers who meet the criteria for access to confidential data. The requests can be sent to Kurdistan University of Medical Sciences Ethics Committee email: ethiccommittee@muk.ac.ir.

4. Is the manuscript presented in an intelligible fashion and written in standard English?

Reviewer #1: Yes

Reviewer #2: Yes

Responses: Thanks for your comment.________________________________________

5. Review Comments to the Author

Reviewer #1: “Iranian Primary Healthcare System’s Response to COVID-19Pandemic

using Healthcare Incident Command System”

Comments on the paper by Arezoo Yari et al

submitted to PLOS ONE (MS number PONE-D-22-30776)

Author : P.N. Lee

Date : 23rd November 2022

While the paper is generally well written and the results clear and well analysed, there are a few points that demand some attention. These are considered below in the order they appear in the paper and not in order of importance.

1. The full title would be better if there were a space between “COVID-19” and “Pandemic” and if the word “the” was inserted after “Response to” and after “using”.

Responses: Thanks for your comment. Based on your comment we made this correction on the title and added the space in the tittle "COVID-19 Pandemic". Also, we inserted the word “the” after “Response to” and after “using”. These changes are highlighted in yellow color in the title.

2. I found the Methods section of the Abstract somewhat confusing at first, as it stated that some DHNs had an active HICS and some did not, so how could data on HICS be collected from those that did not have an active HICS? Later on in the paper it appears that data were in fact collected from all the 60 DHNs. It might also be a good idea to mention the number of DHNs with active and inactive HICS. Perhaps the Method section might read something like the following:

“Methods: The study was performed in Iran in June 2020 in 60 DHNs, 41 of which had an active HICS. Data were collected on eight HCIS constructs from all 60 DHNs by trained crisis management experts to evaluate the effects of HICS use on management of the COVID-19 pandemic”.

Responses: Thanks for your comment. Based on your comment we revised the Methods section of the Abstract. These changes are highlighted in yellow color in the abstract section, page 2, lines 7-10. 

3. The results might better start:

“For all the 60 DHNs, the mean score of the COVID-19 incident command and management was 78.79 ± 11.90 (range 20-100); with mean scores highest for organizational support and cooordination and lowest for logistic and planning”.

Responses: Thanks for your comment. Based on your comment we revised the result section of the Abstract. These changes are highlighted in yellow color in the abstract section, page 2, lines 10-12. 

4. While it is clear enough from Tables 2 and 3 that scores on total incident command and the other seven dimensions were clearly higher in DHNs with an active HICS (p<0.01 in all cases), I wonder whether it is enough to infer from this that HICS should always be active. The study does not investigate health endpoints, and it would be interesting to know, for example, whether the 41 DHNs with active HICS had a lower mortality rate among patients admitted for COVID-19 than did the 19 DHNs without active HICS. Does having active HICS work as regards the virus?

Responses: Thanks for your comment. It is definitely very useful to examine and compare the investigate health endpoints caused by COVID-19 in two groups of DHNs, but it was not the subject of this research, in this study only the management and commanding of the COVID-19 pandemic was investigated in two groups of DHNs.

5. Near the end of the Introduction section, there is a statement that MOHME “necessitated that all health systems use HICS as an essential infrastructure since 2011” and that the following the outbreak of SARS-COV-2 in Iran, “the authorities officially announced the activation of HICS in the healthcare system on all levels”. That being the case, how did it come about that 19 of the 60 DHNs did not use the HICS?

Responses: Thanks for your comment. Based on your comment, we rechecked the instructions and found the following: Although this system was announced in the country's health system in 2011, its use was not mandatory due to some practical problems therefore the Ministry of Health and medical education has been trying to commercialize its use through encouragement. Therefore, we corrected and edited the sentences related to this part, these corrections are highlighted in yellow color in the introduction section, page 5, lines 8-10. 

6. Under “data analysis” it is made clear that the classification of total incident management and command into low, average and high was based on putting the bottom 25% as low, the top 25% as high and the rest as average. But there is nothing about how low, average and high were defined for the seven constructs from organizational support to human resource management. It would be useful to give fuller details as to how these scores were derived, perhaps as a supplementary file which details the questions actually asked and the possible replies.

Responses: Thanks for your comment. Based on your comment we added three files with explanations about how these scores were derived as a supplementary file which details the questions actually asked and the possible replies.

 7. In the Discussion section there is a paragraph starting “In this study, among the studied dimensions, the lowest command scores in the DHNs belonged to the planning dimension”. But according to Table 1, the lowest score is for the logistic dimension.

Responses: Thanks for your comment. Based on your comment we revised this section. these corrections are highlighted in yellow color in the discussion section, page 15, lines 22-23.

8. While the paper is well written, the last paragraph of the introduction could be improved by replacing “didn’t” by “did not”, and by inserting “been” before “considered in our study”.

Responses: Thanks for your comment. Based on your comment we replaced “didn’t” by “did not”, and inserted “been” before “considered in our study”. these corrections are highlighted in yellow color in the last paragraph of the discussion section, page 17, lines 4 and 7.

Reviewer #2: Dear Authors

This is good effort to explain Iranian Primary Healthcare System's Response to COVID-19Pandemic using Healthcare Incident Command System.

It is well written .

But in my opinion it is more useful for local Iranian Journal.

Responses: Thanks for your comment. our study was the evaluation of various dimensions of COVID-19 crisis command based on a global algorithm, which encompassed all the aspects of incident command and management in an emergency. Therefore, the results of this study can be used anywhere in the world. this study is effective in improving our understanding of the role of HICS systems in controlling the COVID-19 pandemic.

6. PLOS authors have the option to publish the peer review history of their article (what does this mean?). If published, this will include your full peer review and any attached files.

Do you want your identity to be public for this peer review? For information about this choice, including consent withdrawal, please see our Privacy Policy.

Reviewer #1: Yes: Peter N Lee

Reviewer #2: No

---

## [Decision Letter · Decision Letter 1]

7 Aug 2023

Iranian primary healthcare system's response to the COVID-19 pandemic using the healthcare incident command system

PONE-D-22-30776R1

Dear Dr. Motlagh,

We’re pleased to inform you that your manuscript has been judged scientifically suitable for publication and will be formally accepted for publication once it meets all outstanding technical requirements.

Kind regards,

Masoud Behzadifar

Academic Editor

PLOS ONE

Additional Editor Comments (optional):

Reviewers' comments:

Reviewer's Responses to Questions

**Comments to the Author**

1. If the authors have adequately addressed your comments raised in a previous round of review and you feel that this manuscript is now acceptable for publication, you may indicate that here to bypass the “Comments to the Author” section, enter your conflict of interest statement in the “Confidential to Editor” section, and submit your "Accept" recommendation.

Reviewer #1: All comments have been addressed

Reviewer #2: (No Response)

2. Is the manuscript technically sound, and do the data support the conclusions?

Reviewer #1: Yes

Reviewer #2: Partly

3. Has the statistical analysis been performed appropriately and rigorously? 

Reviewer #1: Yes

Reviewer #2: I Don't Know

4. Have the authors made all data underlying the findings in their manuscript fully available?

Reviewer #1: (No Response)

Reviewer #2: No

5. Is the manuscript presented in an intelligible fashion and written in standard English?

Reviewer #1: Yes

Reviewer #2: No

6. Review Comments to the Author

Reviewer #1: Thank you for addressing the comments I made on the version of the paper as submitted originally. I have no further comments

Reviewer #2: Dear Authors

You said that your study was the evaluation of various

dimensions of COVID-19 crisis command based on a global algorithm, which

encompassed all the aspects of incident command and management in an emergency.

Therefore, the results of this study can be used anywhere in the world. this study is

effective in improving our understanding of the role of HICS systems in controlling the

COVID-19 pandemic.

But I am not sure about your response.

7. PLOS authors have the option to publish the peer review history of their article (what does this mean?). If published, this will include your full peer review and any attached files.

Reviewer #1: **Yes: **Peter N Lee

Reviewer #2: No

---

## [Editor Report · Acceptance letter]

13 Aug 2023

PONE-D-22-30776R1 

Iranian primary healthcare system's response to the COVID-19 pandemic using the healthcare incident command system 

Dear Dr. Motlagh:

I'm pleased to inform you that your manuscript has been deemed suitable for publication in PLOS ONE. Congratulations! Your manuscript is now with our production department. 

Kind regards, 

on behalf of

Dr. Masoud Behzadifar 

Academic Editor

PLOS ONE